# Provably Efficient Interaction-Grounded Learning with Personalized Reward

**Mengxiao Zhang** *
University of Iowa
mengxiao-zhang@uiowa.edu

**Yuheng Zhang** *
University of Illinois Urbana-Champaign
yuhengz2@illinois.edu

**Haipeng Luo**
University of Southern California
haipengl@usc.edu

**Paul Mineiro**
Microsoft Research
pmineiro@microsoft.com

## Abstract

Interaction-Grounded Learning (IGL) [Xie et al., 2021] is a powerful framework in which a learner aims at maximizing unobservable rewards through interacting with an environment and observing reward-dependent feedback on the taken actions. To deal with personalized rewards that are ubiquitous in applications such as recommendation systems, Maghakian et al. [2022] study a version of IGL with context-dependent feedback, but their algorithm does not come with theoretical guarantees. In this work, we consider the same problem and provide the first provably efficient algorithms with sublinear regret under realizability. Our analysis reveals that the step-function estimator of prior work can deviate uncontrollably due to finite-sample effects. Our solution is a novel Lipschitz reward estimator which underestimates the true reward and enjoys favorable generalization performances. Building on this estimator, we propose two algorithms, one based on explore-then-exploit and the other based on inverse-gap weighting. We apply IGL to learning from image feedback and learning from text feedback, which are reward-free settings that arise in practice. Experimental results showcase the importance of using our Lipschitz reward estimator and the overall effectiveness of our algorithms.

## 1 Introduction

Traditional bandit problems [Auer et al., 2002] or reinforcement learning problems [Sutton and Barto, 2018] assume that the learner has access to the reward, which is her learning goal. However, such explicit reward information is usually difficult to obtain in many real-world scenarios, including human-computer interface applications [Pantic and Rothkrantz, 2003, Freeman et al., 2017] and recommender systems [Yi et al., 2014, Wu et al., 2017]. Recently, Xie et al. [2021] study a new setting called Interaction-Grounded Learning (IGL), where the learner interacts with the environment and receives some feedback on her actions instead of explicit reward signals. The learner needs to discover the implicit information about the reward in the feedback and maximizes the reward.

From an information-theoretic perspective, IGL is intractable unless further assumptions are made. Xie et al. [2021] introduce a conditional independence assumption which states that the feedback is conditionally independent of the action and context given the latent reward. However, this assumption is unrealistic in many scenarios. For example, different users interact with recommender systems in different ways even under the same latent reward [Maghakian et al., 2022]. This inspires us to study IGL with personalized rewards, a setting where the feedback depends on the context. Although

---

*Equal contribution.

Maghakian et al. [2022] study this for recommender systems, they only provide empirical results of their approach, leaving the following question open:

*Can we design provably efficient algorithms for interaction-grounded learning when the feedback depends on the context given the latent reward?*

**Contributions.** In this paper, we answer the question in the positive and provide the first provably efficient algorithms with sublinear regret guarantee for IGL with personalized reward. To achieve this, in Section 3.1, we first propose a novel reward estimator via inverse kinematics. Specifically, using the samples collected by applying a uniform policy, we construct a *Lipschitz reward*, which underestimates the reward for all the policies but approximates the reward for the optimal policy well. With this reward estimator, we propose two algorithms, one based on Explore-then-Exploit (Section 3.2) and the other based on inverse-gap weighting (Section 3.3). Both algorithms achieve $\mathcal{O}(T^{\frac{2}{3}})$ regret. To the best of our knowledge, we are the first to propose provable regret guarantees for IGL with personalized reward. In Section 4, we implement both algorithms and apply them to both an image classification dataset and a conversational dataset. The empirical performance showcases the effectiveness of our algorithm and the importance of using the Lipschitz reward estimator.

## 1.1 Related Work

**Interaction-Grounded Learning (IGL).** Xie et al. [2021] is the first work studying IGL and assumes that the feedback is independent of the context and action conditioned on the latent reward. Xie et al. [2022] further relaxes the assumption to an action-inclusive version, where the feedback is independent of context conditioned on both the action and the reward.[2] Hu et al. [2024] follows the same setting as Xie et al. [2021] but proposes an information-theoretic approach to enforce the conditional independence assumption. However, as we mentioned before, context-dependent feedback is ubiquitous in real-word applications, so such conditional independence assumptions reduce the generality of the IGL framework. Maghakian et al. [2022] is the closest work to ours which also considers personalized rewards. Nonetheless, their work focuses on the empirical side and does not provide any theoretical guarantees.

**Contextual online learning with partial feedback.** Our work is closely related to the recent trend of designing efficient contextual learning algorithms with partial feedback, including contextual bandits [Langford and Zhang, 2007, Agarwal et al., 2012, 2014, Foster and Krishnamurthy, 2018, Foster et al., 2018, Foster and Rakhlin, 2020, Simchi-Levi and Xu, 2021], where the learner receives explicit reward signal of her taken action; contextual bandits with feedback graphs [Zhang et al., 2024a,b], where the learner's observation of the reward is determined based on a feedback graph; and contextual partial monitoring [Bartók and Szepesvári, 2012, Kirschner et al., 2020], where the learner's observation is defined by a signal matrix or a linear observation operator. We adopt and generalize the ideas for designing contextual bandits algorithms [Foster and Rakhlin, 2020] to design algorithms for IGL.

## 2 Preliminary

**Problem setup.** Throughout the paper, we use $[N]$ to denote the set $\{1, 2, \ldots, N\}$ for a positive integer $N$. The problem of online Interative-Grounded Learning (IGL) with personalized reward we consider is defined as follows. The interaction between the learner and the environment lasts for $T$ rounds. At each round $t \in [T]$, the environment reveals a stochastic context $x_t \in \mathcal{X}$ i.i.d. drawn from an unknown distribution $\mathcal{D}$, and the learner decides an action $a_t \in [K]$ from a finite action set of size $K$ based on this context. Then, different from the classic contextual bandits in which the learner observes the binary reward of her chosen action $r(x_t, a_t) \in \{0, 1\}$, she receives feedback $y_t \in \mathcal{Y}$.

---

[2]We point out that the idea behind Xie et al. [2022] cannot be used in our setting where feedback can be dependent on the context. Specifically, Xie et al. [2022] propose an algorithm which learns the value function and the reward decoder separately for each action. Generalizing their idea to our setting would then mean learning the value function and the reward decoder for each context, which is infeasible since there could be infinitely many contexts.

**Feedback dependence assumption.** We follow the assumption proposed in [Maghakian et al., 2022] that the feedback is conditionally independent of the action given the context and the realized reward.

**Assumption 1.** *For arbitrary $(x, a, r, y)$ tuple where the reward $r$ and the feedback $y$ are generated based on the context $x$ and action $a$, we assume $y$ is conditionally independent of action $a$ given context $x$ and reward $r$. In other words, we assume that $y \perp a \mid r, x$.*

As mentioned, compared to prior work [Xie et al., 2021, 2022], this better captures many real-world applications such as recommender systems where different users interact with them in different ways [Beel et al., 2013, Shin, 2020, Maghakian et al., 2022].

**Realizability.** We assume that the learner has access to two function classes $\mathcal{F} \subseteq (\mathcal{X} \times [K] \mapsto [0, 1])$ and $\Phi \subseteq (\mathcal{X} \times \mathcal{Y} \mapsto \{0, 1\})$, where $\mathcal{F}$ characterizes the mean of the reward for a given context-action pair, and $\Phi$ characterizes the realized reward given the context and the received feedback. A policy $\pi : \mathcal{X} \to \Delta(K)$ specifies the action probability conditioned on the context. For each $f \in \mathcal{F}$, we use $\pi_f$ to denote the induced policy which takes action greedily according to $f$, that is, $\pi_f(a|x) = \mathbb{1}\{a = \arg\max_{a' \in [K]} f(x, a')\}, \forall a \in [K]$. We also use the shorthand $\pi^\star$ for the optimal policy $\pi_{f^\star}$. We then make the following realizability assumption following previous contextual bandits literature [Agarwal et al., 2012, Foster et al., 2018, Foster and Rakhlin, 2020, Simchi-Levi and Xu, 2021].

**Assumption 2** (Realizability). *There exists a regression function $f^\star \in \mathcal{F}$ such that $\mathbb{E}[r(x_t, a)|x_t] = f^\star(x_t, a)$ for all $a \in [K]$ and $t \in [T]$. Furthermore, there exists a feedback decoder $\phi^\star \in \Phi$ such that $\phi^\star(x_t, y_t) = r(x_t, a_t)$ for all $t \in [T]$.*

For simplicity, we also assume that $\mathcal{F}$ and $\Phi$ are finite with cardinality $|\mathcal{F}|$ and $|\Phi|$. Our results can be further generalized to broader function classes, which will be discussed in later sections.

**Identifiability.** As mentioned in [Xie et al., 2021, 2022, Maghakian et al., 2022], it is impossible to learn if we do not break the symmetry between reward being $1$ and being $0$. Following [Maghakian et al., 2022], we make the following assumption:

**Assumption 3** (Identifiability). *For any $x \in \mathcal{X}$, $f^\star$ defined in Assumption 2 satisfies that: 1) $\sum_{a=1}^{K} f^\star(x, a) \leq \alpha$ for some $0 < \alpha < \frac{K}{2}$; 2) $\max_{a \in [K]} f^\star(x, a) \geq \theta$ where $\theta > \frac{\alpha}{K - \alpha}$.*

The first condition says that the sum of the expected reward over actions is less than $\frac{K}{2}$ given any context $x$. That is to say, the reward vector is sparse if $f^\star(x, a) \in \{0, 1\}$. The second condition says that for each context $x \in \mathcal{X}$, there exists an action that achieves a large enough expected reward. These two conditions are indeed satisfied by many real-world applications, including the $s$-multi-label classification problem (with $s < K/2$) where $\alpha = s$ and $\theta = 1$.

**Regret.** The learner's performance is measured via the notion of regret, which is defined as the expected difference between the learner's total reward and the one received by the optimal policy:

$$\mathbf{Reg}_{\mathrm{CB}} = \mathbb{E}\left[\sum_{t=1}^{T} f^\star(x_t, \pi^\star(x_t)) - \sum_{t=1}^{T} f^\star(x_t, a_t)\right],$$

**Other notations.** We denote the $(K-1)$-dimensional simplex by $\Delta_K$. Let $\mathbf{1}$ be the all-one vector in an appropriate dimension and $\pi_{\mathrm{Unif}} = \frac{1}{K} \cdot \mathbf{1} \in \Delta_K$. For a $d$-dimensional vector $v \in \mathbb{R}^d$, we denote its $i$-th entry by $v_i$. $\mathbb{1}\{\cdot\}$ is the indicator function and $e_i$ is the $i$-th standard basis vector in an appropriate dimension.

## 3 Methodology

In this section, we discuss our methodology. In Section 3.1, we start from introducing how we construct a Lipschitz reward estimator based on uniform samples, which serves as the key for our algorithm construction. We prove that this estimator is an *underestimator* of the reward, and more importantly matches the reward for the optimal policy. Based on this estimator, we design two algorithms for this problem in Section 3.2 and Section 3.3, with the first one based on explore-then-exploit and the second one based on inverse-gap weighting (IGW).

### 3.1 Reward Estimator Construction via Uniform Exploration

We first show how we construct a reward estimator based on feedback collected from a uniform policy, which serves as the most important component in our two algorithms.

#### 3.1.1 Inverse Kinematics

When the reward for each context-action pair is *deterministic* and binary, Maghakian et al. [2022] show that if the learner uniformly samples an action for any context and is able to accurately predict the posterior probability of her chosen action given the context and the feedback (inverse kinematics), then she is able to infer that the reward is $1$ if that posterior probability is above certain threshold. Here, we generalize this thresholding reward estimator to the *randomized* binary reward case, and further prove that this estimator correctly models the reward for the optimal policy. To see this, we first prove the following lemma showing the exact posterior distribution over actions if the learner selects an action uniformly randomly. This is also proven in [Maghakian et al., 2022, Eq.(2)] as well, and we include it here for completeness.

**Lemma 1.** *For any context $x \in \mathcal{X}$, suppose that the learner picks a uniformly random action $a \in [K]$. Let $r$ and $y$ be its realized reward and the corresponding feedback. Then, under Assumption 1 and Assumption 2, the posterior distribution of $a$ given the context $x$ and feedback $y$ equals to*

$$\Pr[a|x,y] = \frac{f^\star(x,a) \cdot \phi^\star(x,y)}{\sum_{a'=1}^{K} f^\star(x,a')} + \frac{(1 - f^\star(x,a))(1 - \phi^\star(x,y))}{K - \sum_{a'=1}^{K} f^\star(x,a')}, \tag{1}$$

*where $f^\star$ and $\phi^\star$ are the true expected reward and feedback decoder defined in Assumption 2.*

Now we show how to infer the true reward from this inverse kinematics. Specifically, we show:

**Lemma 2.** *For any context $x \in \mathcal{X}$ and action $a \in [K]$, let $r(x,a)$ be the realized reward and $y$ be the feedback. Let $h^\star(x,y) \in \Delta_K$ be such that for each $a \in [K]$,*

$$h_a^\star(x,y) \triangleq \frac{f^\star(x,a) \cdot \phi^\star(x,y)}{\sum_{a'=1}^{K} f^\star(x,a')} + \frac{(1 - f^\star(x,a))(1 - \phi^\star(x,y))}{K - \sum_{a'=1}^{K} f^\star(x,a')}. \tag{2}$$

*Then we have*

- $r(x,a) = \phi^\star(x,y) \geq \mathbb{1}\{h_a^\star(x,y) \geq \frac{\theta}{\alpha}\}$ *for all $a \in [K]$;*

- $r(x,a) = \phi^\star(x,y) = \mathbb{1}\{h_a^\star(x,y) \geq \frac{\theta}{\alpha}\}$ *for $a = \pi^\star(x)$.*

*Proof.* Note that since $\phi^\star(x,y) \in \{0,1\}$, only one term can be non-zero in Eq. (2). If $h_a^\star(x,y) \geq \frac{\theta}{\alpha}$, where $\theta$ and $\alpha$ are defined in Assumption 3, then the reward $\phi^\star(x,y)$ has to be $1$ since otherwise, we have $\phi^\star(x,y) = 0$ and

$$\frac{\theta}{\alpha} \leq h_a^\star(x,y) = \frac{1 - f^\star(x,a)}{K - \sum_{a'=1}^{K} f^\star(x,a')} \leq \frac{1}{K - \alpha} < \frac{\theta}{\alpha},$$

where the second and the third inequality are due to Assumption 3. This leads to a contradiction. Therefore, we know that the realized reward is $1$ if $h_a^\star(x,y)$ is no less than $\frac{\theta}{\alpha}$. Therefore, $\mathbb{1}\{h_a^\star(x,y) \geq \frac{\theta}{\alpha}\}$ can be viewed as an underestimator of $r(x,a)$. To prove the second property, consider the case in which $a = \pi^\star(x)$. Then, we know that when $\phi^\star(x,y) = 1$, we must also have $h_a^\star(x,y) \geq \frac{\theta}{\alpha}$ since

$$h_a^\star(x,y) = \frac{f^\star(x,\pi^\star(x))}{\sum_{i=1}^{K} f^\star(x,a)} \geq \frac{f^\star(x,\pi^\star(x))}{\alpha} \geq \frac{\theta}{\alpha},$$

where the first inequality uses the first property in Assumption 3 and the second inequality uses the second property in Assumption 3. $\qquad\square$

Lemma 2 shows that the function $h^\star(x,y)$ defined in Eq. (2) satisfies two important properties. First, $\mathbb{1}\{h_a^\star(x,y) \geq \frac{\theta}{\alpha}\}$ serves as a reward *underestimator* for all the policies. Second, it matches the reward of the optimal policy. Note that this is different from [Maghakian et al., 2022, Eq.(3)], since

they only consider the *deterministic* binary reward for each context-action pair, and they do not show that the constructed reward estimator matches the reward of the optimal policy.

Based on these two properties, we know that if we have access to $h^\star$, then the policy $\pi$ that maximizes the surrogate reward $\mathbb{E}\left[\mathbb{1}\{h^\star_{\pi(x)}(x,y)\} \geq \frac{\theta}{\alpha}\right]$ also maximizes the true expected reward.

### 3.1.2 Learning the Posterior Distribution via ERM

Next, we show how to learn this $h^\star$ via uniformly collected samples. Define the function class $\mathcal{H}$ as:

$$\mathcal{H} = \left\{ h : \mathcal{X} \times \mathcal{Y} \mapsto \Delta_K, h_a(x,y) = \frac{f(x,a)\phi(x,y)}{\sum_{i=1}^{K} f(x,i)} + \frac{(1-f(x,a))(1-\phi(x,y))}{K - \sum_{i=1}^{K} f(x,i)} \right.$$
$$\left. \Big| f \in \mathcal{F}, \phi \in \Phi, a \in [K] \right\}.$$

Note that $|\mathcal{H}| = |\mathcal{F}| \cdot |\Phi|$. Since $h^\star$ models the posterior distribution over actions when the learner selects her action uniformly randomly, we collect $N$ tuples of $(x_t, a_t, y_t)$ by sampling $a_t$ uniformly from $[K]$ and find the empirical risk minimizer (ERM) $\widehat{h} \in \mathcal{H}$ over these samples using squared loss. In the following lemma, we show that $\widehat{h}$ enjoys $\mathcal{O}\left(\frac{\log|\mathcal{H}|}{N}\right)$ excess risk with high probability.

**Lemma 3.** *Let $\{(x_i, a_i, y_i)\}_{i=1}^N$ be $N$ i.i.d. samples where $x_i \in \mathcal{D}$, $a \in \pi_{\mathrm{Unif}}$, and $y_i$ is the corresponding feedback. Let $\widehat{h}$ be the ERM with respect to the squared loss defined as follows:*

$$\widehat{h} = \operatorname*{argmin}_{h \in \mathcal{H}} \left\{ \sum_{i=1}^{N} \|h(x_i, y_i) - e_{a_i}\|_2^2 \right\}. \tag{3}$$

*Then, with probability at least $1 - \delta$, we have*

$$\mathbb{E}_{x \sim \mathcal{D}, a \sim \pi_{\mathrm{Unif}}, y|x,a}\left[\|\widehat{h}(x,y) - e_a\|_2^2 - \|h^\star(x,y) - e_a\|_2^2\right] \leq \mathcal{O}\left(\frac{\log\frac{|\mathcal{H}|}{\delta}}{N}\right), \tag{4}$$

$$\mathbb{E}_{x \sim \mathcal{D}, a' \sim \pi_{\mathrm{Unif}}, y|x,a'}\left[\|\widehat{h}(x,y) - h^\star(x,y)\|_2\right] \leq \mathcal{O}\left(\sqrt{\frac{\log\frac{|\mathcal{H}|}{\delta}}{N}}\right). \tag{5}$$

The full proof is deferred to Appendix A. Different from the classic one-dimensional squared loss, here we consider the $\ell_2$-loss between two vectors in $\Delta_K$. Directly applying the generalization bound for each entry leads to a $K$-factor worse bound. Instead, our proof is based on the observation that the loss function $\|h(x,y) - e_a\|_2^2$, when seen as a function of $h$, satisfies the so-called strong 1-central condition [Van Erven et al., 2015, Definition 7]. Moreover, these results can be extended to function classes with infinite size and bounded covering number based on Theorem 7.7 of [Van Erven et al., 2015].

### 3.1.3 Constructing Lipschitz Reward Estimators Based on $\widehat{h}$

Now we show how to construct a reward estimator based on the ERM $\widehat{h}$. According to Section 3.1.1, an intuitive form of the reward (under)estimator is $\mathbb{1}\{\widehat{h}_a(x,y) \geq \frac{\theta}{\alpha}\}$. However, since the indicator function is not Lipschitz, $\mathbb{1}\{\widehat{h}_a(x,y) \geq \frac{\theta}{\alpha}\}$ can be very different from $\mathbb{1}\{h^\star_a(x,y) \geq \frac{\theta}{\alpha}\}$ even with the generalization bound proven in Lemma 3. To resolve this issue, we propose a Lipschitz variant of the indicator function (defined in Eq. (6)) and show that it also satisfies the two properties shown in Lemma 2.

**Lemma 4.** *Define $G(v, \beta, \sigma)$ as*

$$G(v, \beta, \sigma) = \frac{1}{\sigma}(v - \beta)\mathbb{1}\{\beta \leq v < \beta + \sigma\} + \mathbb{1}\{v \geq \beta + \sigma\}. \tag{6}$$

*Then, for any context $x \in \mathcal{X}$, action $a \in [K]$, and feedback $y \in \mathcal{Y}$ generated via context $x$ and the realized reward $r(x,a)$, we have the following two properties with $\sigma \triangleq \frac{1}{2}\left(\frac{\theta}{\alpha} - \frac{1}{K-\alpha}\right) > 0$.*

---

**Algorithm 1** Off-Policy IGL

---

Input: number of exploration samples $N$, parameters $\alpha, \theta$ and $\sigma = \frac{1}{2}\left(\frac{\theta}{\alpha} - \frac{1}{K-\alpha}\right)$.

**for** $t = 1, 2, \cdots, 2N$ **do**
  |   Receive context $x_t$, sample $a_t \sim \pi_{\text{Unif}}$, and observe signal $y_t$.
Calculate the empirical risk minimizer $\widehat{h}$ as in Eq. (3).
Construct the reward decoder $\widehat{r}_\sigma(x, y, a) = G(\widehat{h}_a(x, y), \frac{\theta}{\alpha} - \sigma, \sigma)$ where $G$ is defined in Eq. (6).
Calculate $\widehat{\pi} = \text{argmax}_{\pi \in \Pi}\left\{\sum_{i=N+1}^{2N} \pi(a_i | x_i) \cdot \widehat{r}_\sigma(x_i, y_i, a_i)\right\}$ where $\Pi = \{\pi_f : f \in \mathcal{F}\}$.
**for** $t = 2N + 1, \ldots, T$ **do**
  |   Execute policy $\widehat{\pi}$.

---

- $r(x, a) = \phi^\star(x, y) \geq G(h_a^\star(x, y), \frac{\theta}{\alpha} - \sigma, \sigma)$,

- $r(x, a) = \phi^\star(x, y) = G(h_a^\star(x, y), \frac{\theta}{\alpha} - \sigma, \sigma)$ if $a = \pi^\star(x)$,

The proof shares a similar spirit to Lemma 2 and is deferred to Appendix A. Notably, the function $G(v, \beta, \sigma)$ is $\frac{1}{\sigma}$-Lipschitz in $v$, which is important in order to show concentration between the reward estimator with respect to the true posterior distribution $G(h_a^\star(x, y), \frac{\theta}{\alpha} - \sigma, \sigma)$ and that constructed via the ERM function $\widehat{h}$: $G(\widehat{h}_a(x, y), \frac{\theta}{\alpha} - \sigma, \sigma)$. In the following, we will show how to use the reward estimator $G(\widehat{h}_a(x, y), \frac{\theta}{\alpha} - \sigma, \sigma)$ to design algorithms with provable guarantees.

## 3.2 Off-Policy Algorithm

Built on the reward estimator in Section 3.1, we present our off-policy algorithm, summarized in Algorithm 1. Our algorithm follows the explore-then-exploit idea and consists of two phases. In the exploration phase, we perform uniform exploration and collect the dataset $\{x_i, a_i, y_i\}_{i=1}^{2N}$. The first $N$ samples are used to learn the reward estimator $\widehat{r}$ and the rest $N$ samples are used to learn the policy $\widehat{\pi}$ with $\widehat{r}$. For the exploitation phase, we employ the learned policy $\widehat{\pi}$ in the remaining $T - 2N$ iterations. We present the regret bound of Algorithm 1 in the following theorem.

**Theorem 1.** *Under Assumptions 1-3, Algorithm 1 with $N = T^{2/3} K^{2/3} \sigma^{-2/3} \log^{1/3}(|\mathcal{H}|T)$ guarantees that* $\mathbf{Reg}_{\mathsf{CB}} \leq \mathcal{O}\left(T^{2/3} K^{2/3} \sigma^{-2/3} \log^{1/3}(|\mathcal{H}|T)\right)$.

We remark that this is the first provably efficient algorithm for IGL with personalized reward. The key of the proof is to show that the learned policy $\widehat{\pi}$ is near-optimal under the true reward. This is achieved by using the properties of function $G$ in Lemma 4 and proving that the reward decoder $\widehat{r}_\sigma$ is close to the ground-truth. The full proof is deferred to Appendix B. Besides the dependence on $T$, $K$ and $\log |\mathcal{H}|$, our regret bound also depends on $\sigma^{-1}$, which measures how sparse the reward vector is and characterizes the difficulty of the problem. For example, $\sigma^{-1} = \mathcal{O}(1)$ in the multi-class classification problem and $\sigma^{-1} \leq \mathcal{O}(s^2)$ in the $s$-multi-label classification problem.

## 3.3 On-Policy Algorithm

Since on-policy algorithms are more favorable in practice, we further introduce an on-policy algorithm based on the inverse-gap weighting strategy. Following [Foster and Rakhlin, 2020], we assume access to an online regression oracle AlgSq: at each round $t \in [T]$, the oracle AlgSq produces an estimator $\widehat{f}_t$ in the convex hull of $\mathcal{F}$, then receives a context-action-reward tuple $(x_t, a_t, r_t)$. The squared loss of the oracle for this round is defined as $(\widehat{f}_t(x_t, a_t) - r_t)^2$, which is on average assumed to be close to that of the best predictor in $\mathcal{F}$.

**Assumption 4** (Bounded squared loss regret). *For any sequence $\{(x_t, a_t, r_t)\}_{t=1}^T$, the regression oracle AlgSq guarantees the following regret bound for some $\mathbf{Reg}_{\mathsf{Sq}}$ that depends on $T$, $K$, and $\mathcal{F}$:*

$$\sum_{t=1}^T \left(\widehat{f}_t(x_t, a_t) - r_t\right)^2 - \inf_{f \in \mathcal{F}} \sum_{t=1}^T (f(x_t, a_t) - r_t)^2 \leq \mathbf{Reg}_{\mathsf{Sq}}.$$

---

**Algorithm 2** On-policy IGL

---

Input: online regression oracle AlgSq, number of exploration samples $N$, parameters $\alpha, \theta, \gamma$ and $\sigma = \frac{1}{2}\left(\frac{\theta}{\alpha} - \frac{1}{K-\alpha}\right)$.

**for** $t = 1, 2, \cdots, N$ **do**
  |   Receive context $x_t$, sample $a_t \sim \pi_{\text{Unif}}$, and observe signal $y_t$.
Calculate the empirical risk minimizer $\widehat{h}$ as in Eq. (3).
**for** $t = N+1, \ldots, T$ **do**
  |   Receive context $x_t$.
  |   Obtain an estimator $\widehat{f}_t$ from the oracle AlgSq.
  |   Calculate the action distribution $p_t$ as

$$p_{t,a} = \begin{cases} \frac{1}{K + \gamma(\widehat{f}_t(x_t, \widehat{a}) - \widehat{f}_t(x_t, a))}, & a \neq \widehat{a}, \\ 1 - \sum_{a' \neq \widehat{a}} p_{a'}, & a = \widehat{a} \end{cases}, \tag{7}$$

  |   where $\widehat{a} = \text{argmax}_{a \in [K]} \widehat{f}_t(x_t, a)$.
  |   Sample $a_t$ from $p_t$ and receive feedback $y_t$.
  |   Feed $(x_t, a_t, G(\widehat{h}_{a_t}(x_t, y_t), \frac{\theta}{\alpha} - \sigma, \sigma))$ to the oracle AlgSq where $G$ is defined in Eq. (6).

---

Based on the ground-truth inverse kinematics function $h^\star$, we define a function $\underline{f}^\star(x, a) := \mathbb{E}_{y|x,a}\left[G(h^\star_a(x, y), \frac{\theta}{\alpha} - \sigma, \sigma)\right]$, which is always a lower bound on $f^\star(x, a)$ according to Lemma 4. Since we only feed the surrogate reward to the oracle, we make another mild assumption that our regression function class $\mathcal{F}$ also realizes $\underline{f}^\star$.

**Assumption 5** (Lower Bound Realizability). *We also assume that $\underline{f}^\star \in \mathcal{F}$, where $\underline{f}^\star$ is defined as* $\underline{f}^\star(x, a) = \mathbb{E}_{y|x,a}\left[G(h^\star_a(x, y), \frac{\theta}{\alpha} - \sigma, \sigma)\right]$

We now summarize our algorithm in Algorithm 2. After obtaining the inverse kinematics predictor $\widehat{h}$ in the same manner as Algorithm 1, instead of uniform exploring, we use the estimated reward from the oracle and an inverse-gap weighting strategy [Abe and Long, 1999, Foster and Rakhlin, 2020] defined in Eq. (7). Different from the contextual bandit problem where the true reward is given and fed to the oracle, we feed the predicted reward $G(\widehat{h}_{a_t}(x_t, y_t), \frac{\theta}{\alpha} - \sigma, \sigma)$ to the oracle.

One might wonder how such misspecification in rewards would affect the regret bound. Our key observation is that since we use the uniform policy to collect data used to train $\widehat{h}$, the generalization error of $\widehat{h}$ is small for any $a$ due to good coverage of the dataset on each action. Based on this observation, we prove the following theorem for Algorithm 2.

**Theorem 2.** *Under Assumptions 1-5, Algorithm 2 with certain choice of $N$ and $\gamma$ guarantees that* $\mathbf{Reg}_{\mathsf{CB}} = \mathcal{O}\left(\sqrt{KT\mathbf{Reg}_{\mathsf{Sq}}} + \sigma^{-2/3}(KT)^{2/3}\log^{1/3}(|\mathcal{H}|T)\right).$

The proof mainly relies on the generalization bounds in Lemma 3 and the property of $G$ in Lemma 4, and is deferred to Appendix C. We observe that Algorithm 2 enjoys the same dependence on $T$, $K$, $\sigma^{-1}$ as Algorithm 1. For finite $\mathcal{F}$, we can use Vovk's aggregation algorithm [Vovk, 1995] as the regression oracle and achieve $\mathbf{Reg}_{\mathsf{Sq}} = \mathcal{O}(\log|\mathcal{F}|)$, making the second term in the regret bound negligible.[3] While in theory our on-policy algorithm does not seem to be more favorable than the off-policy algorithm (mostly because they both need to uniformly explore for a certain period in order to build $\widehat{h}$), it can still perform better in practice, as shown in our experiments.

## 4 Experiments

In this section, we apply IGL to learning from image feedback and learning from text feedback. Specifically, we conduct experiments on the MNIST dataset and a conversational dataset to verify the effectiveness of our algorithms and the Lipschitz reward estimator constructed by Eq. (6).

---

[3]We refer readers to Foster and Rakhlin [2020] for more examples of regression oracles when $\mathcal{F}$ is not finite.

Table 1: Performance of Algorithm 1 and Algorithm 2 on the MNIST dataset.

| Algorithm | Reward Estimator | Average Progressive Reward | Test Accuracy |
|---|---|---|---|
| Off-policy Algorithm 1 | Binary | 0.614 (0.012) | 72.6% (1.5%) |
| | Lipschitz | 0.638 (0.042) | 75.9% (4.9%) |
| On-policy Algorithm 2 | Binary | 0.718 (0.006) | 89.4% (4.0%) |
| | Lipschitz | 0.740 (0.022) | 90.3% (3.7%) |

## 4.1 Experiments on Learning from Image Feedback

**Experiment Setup**  We first conduct experiments on MNIST dataset and implement both the off-policy algorithm Algorithm 1 and the on-policy algorithm Algorithm 2. The setup is as follows: at each step $t$, the learner receives an image $x_t$ with ground-truth label $l_t$ and picks action $a_t$ from $\{0, \cdots, 9\}$ as the predicted label. If the prediction $a_t$ is correct, the learner receives as feedback an image $y_t$ with digit $(l_t + 1) \mod 10$; otherwise, the learner receives an image with digit $(l_t - 1) \mod 10$. Therefore, different from the experimental setup in [Xie et al., 2021], our feedback $y_t$ does depend on both the context and the reward. Both function classes $\mathcal{H}$ and $\mathcal{F}$ are implemented as two-layer convolutional neural networks.

We use PyTorch framework [Paszke et al., 2019] and parameter-free optimizers [Orabona and Tommasi, 2017] to learn the reward estimator and the policy. For both algorithms, we set the number of exploration samples $N = 5000$ and pick the parameter $\sigma \in \{0, 0.05, 0.1, 0.15, 0.2, 0.25, 0.3\}$ and $\frac{\theta}{\alpha} \in \{\frac{1}{3} + \frac{\sigma}{2}, \frac{1}{2} + \frac{\sigma}{2}\}$. For the on-policy algorithm, we use a time-varying exploration parameter $\gamma_t = \sqrt{Kt}$ as suggested by Foster and Krishnamurthy [2021]. We run the experiments on one NVIDIA GeForce RTX 2080 Ti.

The interaction lasts for $T = 60000$ rounds. We use two metrics to evaluate the performance of the algorithm, including the average progressive reward during the interaction process and the test accuracy on a held-out test set containing 10000 samples. When evaluating the on-policy algorithm on the test set, we take actions greedily according to $\widehat{f}_T$. Since we use the uniform policy to collect data for learning the reward estimator, the progressive reward at that phase is counted as $\frac{1}{K}$.

**Results**  We run the experiments with 4 different random seeds and report the mean value and standard deviation in Table 1. The running averaged reward over the entire $T$ rounds are shown in Figure 1. We can see that both algorithms achieve good performance, despite never observing any true labels. While the theoretical regret guarantees for both algorithms are of the same order, empirically, the on-policy Algorithm 2 performs better than Algorithm 1 with over 90% test accuracy. This is because the on-policy algorithm uses the inverse-gap weighting strategy to achieve a better trade-off between exploration and exploitation, while the off-policy algorithm learns the policy from uniform exploration. On the other hand, to demonstrate the effectiveness of the Lipschitz reward estimator, we compare the performances of the Lipschitz estimator Eq. (6) with a binary reward estimator $\widehat{r}_{\text{binary}}(x, y, a) = \mathbb{1}\{\widehat{h}_a(x, y) \geq \frac{\theta}{\alpha}\}$, where the parameter $\frac{\theta}{\alpha}$ is searched over the same space. The results in Table 1 show that the Lipschitz reward estimator improves over the binary one by a clear margin for both algorithms. This matches our theoretical analysis that highlights the vital role of the Lipschitzness of the reward estimator in obtaining good regret guarantees. We also plot the performance of Algorithm 2 under both true reward and constructed reward in the left figure of Figure 2.[4] The figure shows that the constructed reward is indeed a lower bound of the true reward, and the policy can learn from the constructed reward effectively.

## 4.2 Experiments on Learning From Text Feedback

We further consider an application of learning with text feedback. The IGL framework fits this problem well since this is a natural reward-free setting involving learning from user's text feedback, which is idiosyncratically related to the quality of the actions taken and the question asked. Specifically, given the input of a question, the learner needs to select one of the $K$ possible answers. Instead of

---

[4]Both the true reward and the constructed reward values are averaged over the rounds after the first $N$ rounds of uniform exploration.

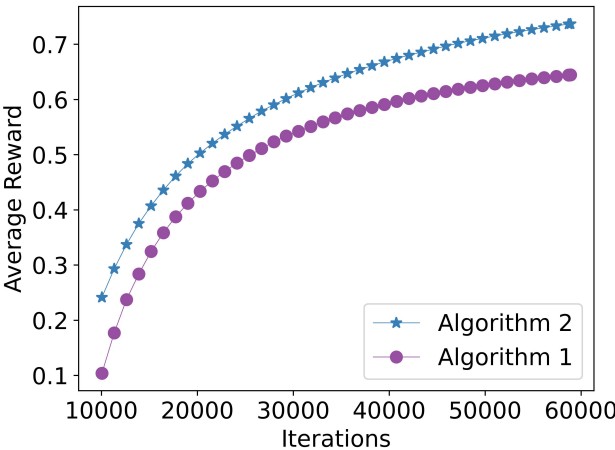

Figure 1: Running averaged reward of Algorithm 1 and Algorithm 2 on MNIST. Note that Algorithm 1 uniformly explores in the first $2N = 10000$ rounds, and thus its averaged reward at $t = 10000$ is about $1/K = 0.1$.

receiving whether the selected answer is correct, the learner only receives user's text feedback to the chosen answer. The goal of the learner is to choose the best answer only based on such text feedback.

**Dataset Construction** Our dataset is constructed as follows. Specifically, we construct our question set $S = \{q_i\}_{i \in [20000]}$ from Chatbot Arena datasets [Zheng et al., 2024]. Then, for each question $q_i \in S$ we ask a larger language model $\mathcal{G}$ with a high ELO score on the chatsys leaderboard [Chiang et al., 2024] to generate a "good" answer $g_{i,0}$ with $r_{i,0} = 1$; and ask a (much) smaller language model $\mathcal{B}$ with a (much) lower ELO score to generate 4 "bad" answers $b_{i,j}$ with reward $r_{i,j} = 0$, $j \in \{1, 2, 3, 4\}$. Specifically, we pick $\mathcal{G}$ to be "Qwen1.5-32B-Chat" with ELO score 1134 and $\mathcal{B}$ to be "Qwen1.5-0.5B-Chat" with ELO score[5] less than 804 [Bai et al., 2023].[6] For each question-answer tuple $(q_i, g_{i,0}, b_{i,1}, b_{i,2}, b_{i,3}, b_{i,4})$, we ask another large language model $\mathcal{R}$ to simulate a user response $f_{i,j}$, $j \in \{0, 1, 2, 3, 4\}$ to the good (bad) answers under the instruction that the user is satisfied (unsatisfied) with the answer. We pick $\mathcal{R}$ to be Qwen1.5-32B-Chat as well. This forms our final dataset $S_{\text{Conv}} = \{(q_i, (g_{i,0}, f_{i,0}, r_{i,0}), \{(b_{i,j}, f_{i,j}, r_{i,j})\}_{j \in [4]})\}_{i \in [20000]}$. Again, the true reward is never revealed to the learner, and we only use this reward to measure the performance of our algorithm. The prompt we use is deferred to Appendix D.1. We generate our dataset using one A100 GPU for two weeks.

### 4.2.1 Algorithm Configurations and Results

Given the superior performance of Algorithm 2 over Algorithm 1 on the MNIST dataset shown in Section 4.1, we only test Algorithm 2 on the conversational dataset. We use the first $N = 10000$ data points to learn $\widehat{h}$ and the remaining $|S| - N = 10000$ data points to learn the optimal policy based on the reward function constructed via $\widehat{h}$. We use the same parameter-free optimizer as the one in Section 4.1. Next, we introduce the construction of $\widehat{h}$ and the regression oracle:

**Inverse kinematic model.** The model class $\mathcal{H}$ we consider is the pretrained language model Llama-3-8B-Instruct [AI@Meta, 2024] with an additional multi-class classification head.[7] The language model is prompted with a question-answer-feedback tuple $(x, a, y)$; see Appendix D.2 for the prompt. To learn the inverse kinematic model, we use parameter efficient fine-tuning [Mangrulkar et al., 2024] with a rank-1 LORA adapter [Hu et al., 2021] and binary cross entropy loss, which is with respect to the indicator of whether-or-not the predicted action corresponds to the action selected in the tuple.

---

[5]The bad model $\mathcal{B}$ is not displayed on the leaderboard; 804 is the lowest displayed score.

[6]"Qwen1.5-32B-Chat" is available at https://huggingface.co/Qwen/Qwen1.5-32B-Chat and "Qwen1.5-0.5B-Chat" is available at https://huggingface.co/Qwen/Qwen1.5-0.5B-Chat.

[7]"Llama-3-8B-Instruct": https://huggingface.co/meta-llama/Meta-Llama-3-8B-Instruct.

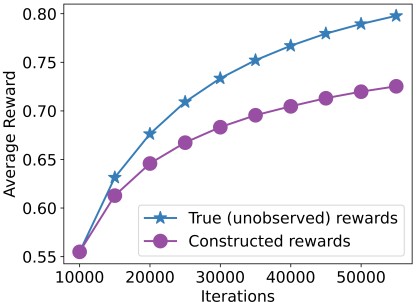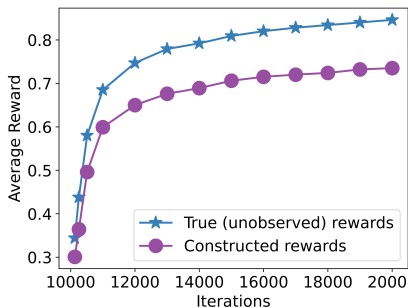

Figure 2: Performance of Algorithm 2 under true (unobserved) rewards and constructed rewards. **Left figure**: Results on MNIST dataset after the first $N$ uniform exploration rounds. **Right figure**: Results on our conversational dataset.

We successfully learn $\widehat{h}$ using one A100 GPU within 6 hours. After obtaining $\widehat{h}$, we construct the reward estimator $G(\widehat{h}_a(x,y), \frac{\theta}{\alpha} - \sigma, \sigma)$ with $\sigma = 0.1$, $\alpha = \frac{K}{2} = \frac{5}{2}$, and $\theta = 1$.

**Regression oracle.** Similar to the inverse kinematic model, the reward prediction function class $\mathcal{F}$ is again based on the pretrained Llama-3-8B-Instruct model but with an additional regression head. Specifically, the language model is prompted only with a question-answer pair $(x, a)$ using the prompt deferred to Appendix D.3 and predicts a score in $[0, 1]$ for this question-answer pair. The regression oracle again applies parameter efficient fine-tuning with a different rank-1 LORA adapter on the regression loss, which measures the error of predicting the output of the reward predictor $G(\widehat{h}_a(x,y), \frac{\theta}{\alpha} - \sigma, \sigma)$. This process is done on one A100 GPU within 3 hours.

**Results.** The empirical results on the conversational dataset are shown in Figure 2. We show the running averaged true reward and the running average constructed reward received by Algorithm 2 after the first $N = 10000$ rounds of learning $\widehat{h}$. The $x$-axis is the time horizon and $y$-axis is the value of average reward. Similar to our experiment results in Section 4.1, the right figure in Figure 2 shows that our constructed reward estimator is a lower bound on the true reward, matching our theoretical results in Lemma 4, and Algorithm 2 is able to learn the reward effectively through the text feedback with the constructed reward estimator.

## Acknowledgments and Disclosure of Funding

HL and MZ were supported by NSF Award IIS-1943607.

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

# A  Proofs in Section 3.1

**Lemma 1.** *For any context $x \in \mathcal{X}$, suppose that the learner picks a uniformly random action $a \in [K]$. Let $r$ and $y$ be its realized reward and the corresponding feedback. Then, under Assumption 1 and Assumption 2, the posterior distribution of $a$ given the context $x$ and feedback $y$ equals to*

$$\Pr[a|x,y] = \frac{f^\star(x,a) \cdot \phi^\star(x,y)}{\sum_{a'=1}^{K} f^\star(x,a')} + \frac{(1 - f^\star(x,a))(1 - \phi^\star(x,y))}{K - \sum_{a'=1}^{K} f^\star(x,a')}, \tag{1}$$

*where $f^\star$ and $\phi^\star$ are the true expected reward and feedback decoder defined in Assumption 2.*

*Proof.*

$$
\begin{aligned}
\Pr[a|x,y] &= \frac{\Pr[a|x] \cdot \Pr[y|x,a]}{\Pr[y|x]} \\
&= \Pr[a|x] \frac{\Pr[r=1|x,a] \cdot \Pr[y|x,a,r=1] + \Pr[r=0|x,a] \cdot \Pr[y|x,a,r=0]}{\Pr[y|x]} \\
&= \Pr[a|x] \frac{f^\star(x,a) \cdot \Pr[y|x,r=1] + (1 - f^\star(x,a)) \cdot \Pr[y|x,r=0]}{\Pr[y|x]} \\
&= \Pr[a|x] \frac{f^\star(x,a) \cdot \Pr[r=1|x,y]}{\Pr[r=1|x]} + \Pr[a|x] \frac{(1 - f^\star(x,a)) \cdot \Pr[r=0|x,y]}{\Pr[r=0|x]} \\
&= \Pr[a|x] \frac{f^\star(x,a) \cdot \Pr[r=1|x,y]}{\sum_{a'=1}^{K} \Pr[a'|x] \Pr[r=1|x,a']} + \Pr[a|x] \frac{(1 - f^\star(x,a)) \cdot \Pr[r=0|x,y]}{\sum_{a'=1}^{K} \Pr[a'|x] \Pr[r=0|x,a']} \\
&= \Pr[a|x] \frac{f^\star(x,a) \cdot \Pr[r=1|x,y]}{\sum_{a'=1}^{K} \Pr[a'|x] f^\star(x,a')} + \Pr[a|x] \frac{(1 - f^\star(x,a)) \cdot \Pr[r=0|x,y]}{\sum_{a'=1}^{K} \Pr[a'|x](1 - f^\star(x,a'))}.
\end{aligned}
$$

Recall that $\pi_{\text{Unif}} = \frac{1}{K} \cdot \mathbf{1}$ is the policy which uniformly samples an action regardless of the context. Under $\pi_{\text{Unif}}$, we know that

$$\Pr[a|x,y] = \frac{f^\star(x,a) \cdot \phi^\star(x,y)}{\sum_{a'=1}^{K} f^\star(x,a')} + \frac{(1 - f^\star(x,a))(1 - \phi^\star(x,y))}{K - \sum_{a'=1}^{K} f^\star(x,a')}.$$

$\square$

**Lemma 3.** *Let $\{(x_i, a_i, y_i)\}_{i=1}^{N}$ be $N$ i.i.d. samples where $x_i \in \mathcal{D}$, $a \in \pi_{\text{Unif}}$, and $y_i$ is the corresponding feedback. Let $\widehat{h}$ be the ERM with respect to the squared loss defined as follows:*

$$\widehat{h} = \operatorname*{argmin}_{h \in \mathcal{H}} \left\{ \sum_{i=1}^{N} \|h(x_i, y_i) - e_{a_i}\|_2^2 \right\}. \tag{3}$$

*Then, with probability at least $1 - \delta$, we have*

$$\mathbb{E}_{x \sim \mathcal{D}, a \sim \pi_{\text{Unif}}, y|x,a} \left[ \|\widehat{h}(x,y) - e_a\|_2^2 - \|h^\star(x,y) - e_a\|_2^2 \right] \leq \mathcal{O}\left( \frac{\log \frac{|\mathcal{H}|}{\delta}}{N} \right), \tag{4}$$

$$\mathbb{E}_{x \sim \mathcal{D}, a' \sim \pi_{\text{Unif}}, y|x,a'} \left[ \|\widehat{h}(x,y) - h^\star(x,y)\|_2 \right] \leq \mathcal{O}\left( \sqrt{\frac{\log \frac{|\mathcal{H}|}{\delta}}{N}} \right). \tag{5}$$

*Proof.* For notational convenience, we denote $(x,y,a)$ by $Z$ and define $\ell_h(Z) = \|h(x,y) - e_a\|_2^2$. Now we aim to show that $\ell_h(Z)$ satisfies the strong $\eta$-central condition for some $\eta > 0$ [Van Erven et al., 2015]. Specifically, we aim to show that

$$\mathbb{E}_Z \left[ \exp(-\eta(\ell_h(Z) - \ell_{h^\star}(Z))) \right] \leq 1. \tag{8}$$

To show this, direct calculation shows that

$$\mathbb{E}_Z \left[ \exp(-\eta(\ell_h(Z) - \ell_{h^\star}(Z))) \right]$$

$$= \mathbb{E}_{x,y} \left[ \exp(-\eta \|h(x,y) - h^\star(x,y)\|_2^2) \cdot \mathbb{E}_{a|x,y} \left[ \exp \left( -2\eta(h(x,y) - h^\star(x,y))^\top (h^\star(x,y) - e_a) \right) \right] \right].$$

Since $\mathbb{E}_{a|x,y}[e_a] = h^\star(x,y)$, the random variable $(h(x,y) - h^\star(x,y))^\top (h^\star(x,y) - e_a)$ given $x$ and $y$ is zero-mean and is within the range $[-2\|h(x,y) - h^\star(x,y)\|_2, 2\|h(x,y) - h^\star(x,y)\|_2]$. Therefore, we know that $(h(x,y) - h^\star(x,y))^\top (h^\star(x,y) - e_a)$ is $\|h(x,y) - h^\star(x,y)\|_2^2$-sub-Gaussian and we have

$$\mathbb{E}_Z \left[ \exp(-\eta(\ell_h(Z) - \ell_{h^\star}(Z))) \right]$$

$$\leq \mathbb{E}_{x,y} \left[ \exp(-\eta \|h(x,y) - h^\star(x,y)\|_2^2) \exp \left( \frac{4\eta^2 \|h(x,y) - h^\star(x,y)\|_2^2}{2} \right) \right]$$

$$= \mathbb{E}_{x,y} \left[ \exp \left( (2\eta^2 - \eta) \|h(x,y) - h^\star(x,y)\|_2^2 \right) \right].$$

Picking $\eta = \frac{1}{2}$ proves Eq. (8). Noticing the fact that $|\ell_h(Z)| \leq 4$ for all $h \in \mathcal{H}$ and applying Theorem 7.6 in [Van Erven et al., 2015] proves Eq. (4).

Now we prove Eq. (5). Noticing the fact that $\mathbb{E}_{a|x,y}[e_a] = h^\star(x,y)$ and applying this to Eq. (4), we know that with probability at least $1 - \delta$,

$$\mathbb{E}_{x,y} \left[ \|\widehat{h}(x,y) - h^\star(x,y)\|_2^2 \right] \leq \mathcal{O} \left( \frac{\log \frac{|\mathcal{H}|}{\delta}}{N} \right).$$

This means that

$$\mathbb{E}_{x \sim \mathcal{D}} \mathbb{E}_{a' \sim \pi_{\text{Unif}}} \mathbb{E}_{y|x,a'} \left[ \|\widehat{h}(x,y) - h^\star(x,y)\|_2^2 \right] \leq \mathcal{O} \left( \frac{\log \frac{|\mathcal{H}|}{\delta}}{N} \right). \tag{9}$$

Further applying Jensen's inequality shows that

$$\mathbb{E}_{x \sim \mathcal{D}} \mathbb{E}_{a' \sim \pi_{\text{Unif}}} \mathbb{E}_{y|x,a'} \left[ \|\widehat{h}(x,y) - h^\star(x,y)\|_2 \right]$$

$$= \mathbb{E}_{x \sim \mathcal{D}} \mathbb{E}_{a' \sim \pi_{\text{Unif}}} \mathbb{E}_{y|x,a'} \left[ \sqrt{\|\widehat{h}(x,y) - h^\star(x,y)\|_2^2} \right]$$

$$\leq \sqrt{\mathbb{E}_{x \sim \mathcal{D}} \mathbb{E}_{a' \sim \pi_{\text{Unif}}} \mathbb{E}_{y|x,a'} \left[ \|\widehat{h}(x,y) - h^\star(x,y)\|_2^2 \right]} \qquad \text{(Jensen's inequality)}$$

$$\leq \mathcal{O} \left( \sqrt{\frac{\log \frac{|\mathcal{H}|}{\delta}}{N}} \right).$$

$\square$

**Lemma 4.** *Define $G(v, \beta, \sigma)$ as*

$$G(v, \beta, \sigma) = \frac{1}{\sigma}(v - \beta) \mathbb{1}\{\beta \leq v < \beta + \sigma\} + \mathbb{1}\{v \geq \beta + \sigma\}. \tag{6}$$

*Then, for any context $x \in \mathcal{X}$, action $a \in [K]$, and feedback $y \in \mathcal{Y}$ generated via context $x$ and the realized reward $r(x,a)$, we have the following two properties with $\sigma \triangleq \frac{1}{2} \left( \frac{\theta}{\alpha} - \frac{1}{K-\alpha} \right) > 0$.*

- $r(x,a) = \phi^\star(x,y) \geq G(h_a^\star(x,y), \frac{\theta}{\alpha} - \sigma, \sigma)$,
- $r(x,a) = \phi^\star(x,y) = G(h_a^\star(x,y), \frac{\theta}{\alpha} - \sigma, \sigma)$ *if $a = \pi^\star(x)$,*

*Proof.* To prove the first property, we show $\phi^\star(x,y) = 1$ if $h_a^\star(x,y) \geq \frac{\theta}{\alpha} - \sigma$. Specifically, if $h_a^\star(x,y) \geq \frac{\theta}{\alpha} - \sigma$ and $\phi^\star(x,y) = 0$, then we know that

$$\frac{\theta}{\alpha} - \sigma \leq h_a^\star(x,y) = \frac{1 - f^\star(x,a)}{K - \sum_{i=1}^K f^\star(x,i)} \leq \frac{1}{K - \alpha},$$

leading to a contradiction according to the definition of $\sigma$. Therefore, when $h_a^\star(x, y) \geq \frac{\theta}{\alpha} - \sigma$, we have $\phi^\star(x, y) = 1$, which is surely an upper bound of $G(h_a^\star(x, y), \frac{\theta}{\alpha} - \sigma, \sigma)$ since $G(h_a^\star(x, y), \frac{\theta}{\alpha} - \sigma, \sigma) \leq 1$. When $h_a^\star(x, y) < \frac{\theta}{\alpha} - \sigma$, by definition, we know that $G(h_a^\star(x, y), \frac{\theta}{\alpha} - \sigma, \sigma) = 0 \leq \phi^\star(x, y)$. Combining the two cases finishes the proof for the first property.

To prove the second property, note that when $\phi^\star(x, y) = 1$ and $a = \pi^\star(x)$, we have

$$h_{\pi^\star(x)}^\star(x, y) = \frac{f^\star(x, \pi^\star(x))}{\sum_{i=1}^K f^\star(x, i)} \geq \frac{\theta}{\alpha},$$

where the last inequality is by Assumption 3. This means that $G(h_a^\star(x, y), \frac{\theta}{\alpha} - \sigma, \sigma) = 1 = \phi^\star(x, y)$. When $\phi^\star(x, y) = 0$, we have

$$h_{\pi^\star(x)}^\star(x, y) = \frac{1 - f^\star(x, \pi^\star(x))}{K - \sum_{i=1}^K f^\star(x, i)} \leq \frac{1}{K - \alpha} \leq \frac{\theta}{\alpha} - \sigma,$$

meaning that $G(h_a^\star(x, y), \frac{\theta}{\alpha} - \sigma, \sigma) = 0$. This finishes the proof. $\qquad\square$

# B    Proofs in Section 3.2

For any policy $\pi$ and reward estimator $r_\sigma$, we define the value function $V(\pi, r_\sigma)$ and the empirical value function $\widehat{V}(\pi, r_\sigma)$ as follows:

$$\widehat{V}(\pi, r_\sigma) = \frac{1}{N} \sum_{i=N+1}^{2N} K \cdot \pi(a_i | x_i) \cdot r_\sigma(x_i, y_i, a_i), \tag{10}$$

$$V(\pi, r_\sigma) = \mathbb{E}_{x \sim \mathcal{D}, a \sim \pi_{\text{Unif}}, y | x, a} \left[ K \cdot \pi(a | x) \cdot r_\sigma(x, y, a) \right], \tag{11}$$

where $\{(x_i, a_i, y_i)\}_{i=N+1}^{2N}$ is collected with the uniform policy $\pi_{\text{Unif}}$. The true value function $V(\pi)$ is defined as:

$$V(\pi) = \mathbb{E}_{x \sim \mathcal{D}} \left[ \sum_{a=1}^K \pi(a | x) \cdot f^\star(x, a) \right].$$

Using Lemma 4, we can establish the equivalence between $V(\pi^\star)$ and $V(\pi^\star, r_\sigma^\star)$ where $r_\sigma^\star(x, y, a) = G(h_a^\star(x, y), \frac{\theta}{\alpha} - \sigma, \sigma)$.

$$\begin{aligned}
V(\pi^\star) &= \mathbb{E}_{x \sim \mathcal{D}, a \sim \pi_{\text{Unif}}} \left[ K \cdot \pi^\star(a | x) \cdot f^\star(x, a) \right] \\
&= \mathbb{E}_{x \sim \mathcal{D}, a \sim \pi_{\text{Unif}}, y | x, a} \left[ K \cdot \pi^\star(a | x) \cdot \phi^\star(x, y) \right] & \text{(Assumption 2)} \\
&= \mathbb{E}_{x \sim \mathcal{D}, a \sim \pi_{\text{Unif}}, y | x, a} \left[ K \cdot \pi^\star(a | x) \cdot G\left( h_a^\star(x, y), \frac{\theta}{\alpha} - \sigma, \sigma \right) \right] & \text{(Lemma 4)} \\
&= V(\pi^\star, r_\sigma^\star). & (12)
\end{aligned}$$

In the following lemma, we prove that $\widehat{\pi}$ is near optimal under the true reward.

**Lemma 5.** *Under Assumption 2 and Assumption 3, the following holds with probability at least $1 - 3\delta$:*

$$V(\pi^\star) - V(\widehat{\pi}) \leq \mathcal{O}\left( \frac{K}{\sigma} \sqrt{\frac{\log \frac{|\mathcal{H}|}{\delta}}{N}} \right),$$

*where $\widehat{\pi}$ is defined in Algorithm 1.*

*Proof.* We first introduce some high probability events that the following analysis is based on. First, according to Hoeffding's inequality together with a union bound over $\pi \in \Pi$ (note that $|\Pi| = |\mathcal{F}| \leq |\mathcal{H}|$), with probability at least $1 - \delta$, we have for any $\pi \in \Pi$

$$\left| V(\pi, r_\sigma^\star) - \widehat{V}(\pi, r_\sigma^\star) \right| \leq \mathcal{O}\left( K \cdot \sqrt{\frac{\log \frac{|\mathcal{H}|}{\delta}}{N}} \right). \tag{13}$$

In addition, according to Hoeffding's inequality with a union bound over $h \in \mathcal{H}$, with probability at least $1 - \delta$, we have for any $h \in \mathcal{H}$, the following inequality holds:

$$\left| \frac{1}{N} \sum_{i=N+1}^{2N} \|h(x_i, y_i) - h^\star(x_i, y_i)\|_2 - \mathbb{E}_{x \sim \mathcal{D}, a \sim \pi_{\text{Unif}}, y|x, a} \left[ \|h(x, y) - h^\star(x, y)\|_2 \right] \right|$$

$$\leq \mathcal{O}\left( \sqrt{\frac{\log \frac{|\mathcal{H}|}{\delta}}{N}} \right). \tag{14}$$

Combining Eq. (14) with Eq. (5) in Lemma 3, we know that with probability at least $1 - 2\delta$,

$$\frac{1}{N} \sum_{i=N+1}^{2N} \left| \widehat{h}_{a_i}(x_i, y_i) - h^\star_{a_i}(x_i, y_i) \right| \leq \frac{1}{N} \sum_{i=N+1}^{2N} \|\widehat{h}(x_i, y_i) - h^\star(x_i, y_i)\|_2$$

$$\leq \mathcal{O}\left( \sqrt{\frac{\log \frac{|\mathcal{H}|}{\delta}}{N}} \right). \tag{15}$$

The following analysis is based on the condition that Eq. (13) and Eq. (15) hold.

**Bounding** $\left| \widehat{V}(\pi, \widehat{r}_\sigma) - \widehat{V}(\pi, r^\star_\sigma) \right|$. We first show that for any policy $\pi \in \Pi$, the prediction from $\widehat{r}_\sigma$ is close to $r^\star_\sigma$ in terms of the empirical value function defined in Eq. (10), where $\widehat{r}_\sigma(x, y, a) \triangleq G(\widehat{h}_a(x, y), \frac{\theta}{\alpha} - \sigma, \sigma)$.

$$\left| \widehat{V}(\pi, \widehat{r}_\sigma) - \widehat{V}(\pi, r^\star_\sigma) \right| \leq \frac{1}{N} \sum_{i=N+1}^{2N} K \cdot \pi(a_i | x_i) \left| \widehat{r}_\sigma(x_i, y_i, a_i) - r^\star_\sigma(x_i, y_i, a_i) \right|$$

$$\leq \frac{K}{N} \sum_{i=N+1}^{2N} \left| \widehat{r}_\sigma(x_i, y_i, a_i) - r^\star_\sigma(x_i, y_i, a_i) \right|$$

$$= \frac{K}{N} \sum_{i=N+1}^{2N} \left| G(\widehat{h}_{a_i}(x_i, y_i), \frac{\theta}{\alpha} - \sigma, \sigma) - G(h^\star_{a_i}(x_i, y_i), \frac{\theta}{\alpha} - \sigma, \sigma) \right|$$

$$\leq \frac{K}{N\sigma} \sum_{i=N+1}^{2N} \left| \widehat{h}_{a_i}(x_i, y_i) - h^\star_{a_i}(x_i, y_i) \right|$$

$$\leq \frac{K}{N\sigma} \sum_{i=N+1}^{2N} \|\widehat{h}(x_i, y_i) - h^\star(x_i, y_i)\|_2$$

$$\leq \mathcal{O}\left( \frac{K}{\sigma} \sqrt{\frac{\log \frac{|\mathcal{H}|}{\delta}}{N}} \right). \tag{16}$$

The third inequality is because $G(v, \beta, \sigma)$ is $\frac{1}{\sigma}$-Lipschitz in $v$ and the last inequality is from Eq. (15).

**Lower bound** $V(\pi)$ **by** $\widehat{V}(\pi, \widehat{r}_\sigma)$. Then, we show that for any policy $\pi \in \Pi$, $V(\pi)$ is lower bounded by $\widehat{V}(\pi, \widehat{r}_\sigma)$:

$$V(\pi) = \mathbb{E}_{x \sim \mathcal{D}} \left[ \sum_{a=1}^{K} \pi(a | x) f^\star(x, a) \right]$$

$$= \mathbb{E}_{x \sim \mathcal{D}, a \sim \pi_{\text{Unif}}} \left[ K \pi(a | x) f^\star(x, a) \right]$$

$$= \mathbb{E}_{x \sim \mathcal{D}, a \sim \pi_{\text{Unif}}} \mathbb{E}_{y | x, a} \left[ K \pi(a | x) \phi^\star(x, y) \right] \qquad \text{(Assumption 2)}$$

$$\geq \mathbb{E}_{x \sim \mathcal{D}, a \sim \pi_{\text{Unif}}} \mathbb{E}_{y | x, a} \left[ K \pi(a | x) G(h^\star_a(x, y), \frac{\theta}{\alpha} - \sigma, \sigma) \right]$$

$$= V(\pi, r_\sigma^\star) \tag{Eq. (11)}$$

$$= V(\pi, r_\sigma^\star) - \widehat{V}(\pi, r_\sigma^\star) + \widehat{V}(\pi, r_\sigma^\star) - \widehat{V}(\pi, \widehat{r}_\sigma) + \widehat{V}(\pi, \widehat{r}_\sigma)$$

$$\geq \widehat{V}(\pi, \widehat{r}_\sigma) - \mathcal{O}\left( \frac{K}{\sigma} \sqrt{\frac{\log \frac{|\mathcal{H}|}{\delta}}{N}} \right). \tag{17}$$

The first inequality is from Lemma 4 and the second inequality is from Eq. (13) and Eq. (16). Recall that $\widehat{\pi} = \max_{\pi \in \Pi} \widehat{V}(\pi, \widehat{r}_\sigma)$, we then know that with probability at least $1 - 3\delta$,

$$V(\pi^\star) - V(\widehat{\pi}) \leq V(\pi^\star) - \widehat{V}(\widehat{\pi}, \widehat{r}_\sigma) + \mathcal{O}\left( \frac{K}{\sigma} \sqrt{\frac{\log \frac{|\mathcal{H}|}{\delta}}{N}} \right) \tag{Eq. (17)}$$

$$\leq V(\pi^\star) - \widehat{V}(\pi^\star, \widehat{r}_\sigma) + \mathcal{O}\left( \frac{K}{\sigma} \sqrt{\frac{\log \frac{|\mathcal{H}|}{\delta}}{N}} \right) \qquad \text{(optimality of } \widehat{\pi} \text{ on } \widehat{V}(\pi, \widehat{r}_\sigma))$$

$$\leq V(\pi^\star) - \widehat{V}(\pi^\star, r_\sigma^\star) + \mathcal{O}\left( \frac{K}{\sigma} \sqrt{\frac{\log \frac{|\mathcal{H}|}{\delta}}{N}} \right) \tag{Eq. (16)}$$

$$\leq V(\pi^\star) - V(\pi^\star, r_\sigma^\star) + \mathcal{O}\left( \frac{K}{\sigma} \sqrt{\frac{\log \frac{|\mathcal{H}|}{\delta}}{N}} \right) \tag{Eq. (13)}$$

$$= \mathcal{O}\left( \frac{K}{\sigma} \sqrt{\frac{\log \frac{|\mathcal{H}|}{\delta}}{N}} \right), \tag{Eq. (12)}$$

which finishes the proof. $\qquad \square$

Next, we prove Theorem 1.

**Theorem 1.** *Under Assumptions 1-3, Algorithm 1 with $N = T^{2/3} K^{2/3} \sigma^{-2/3} \log^{1/3}(|\mathcal{H}|T)$ guarantees that $\mathbf{Reg}_{\mathsf{CB}} \leq \mathcal{O}\left( T^{2/3} K^{2/3} \sigma^{-2/3} \log^{1/3}(|\mathcal{H}|T) \right)$.*

*Proof.* We bound $\mathbf{Reg}_{\mathsf{CB}}$ as follows:

$$\mathbf{Reg}_{\mathsf{CB}} = \mathbb{E}\left[ \sum_{t=1}^{T} f^\star(x_t, \pi^\star(x_t)) - \sum_{t=1}^{T} f^\star(x_t, a_t) \right]$$

$$\leq 2N + \mathbb{E}\left[ \sum_{t=2N+1}^{T} f^\star(x_t, \pi^\star(x_t)) - \sum_{t=2N+1}^{T} f^\star(x_t, a_t) \right]$$

$$= 2N + \mathbb{E}\left[ \sum_{t=2N+1}^{T} f^\star(x_t, \pi^\star(x_t)) - \sum_{t=2N+1}^{T} f^\star(x_t, \widehat{\pi}(x_t)) \right]$$

$$\leq 2N + T \cdot \mathbb{E}\left[ V(\pi^\star) - V(\widehat{\pi}) \right]$$

$$\leq \mathcal{O}\left( N + \frac{TK}{\sigma} \sqrt{\frac{\log(|\mathcal{H}|T)}{N}} \right).$$

The last step is from Lemma 5 with $\delta = \frac{1}{T}$. Picking $N = T^{2/3} K^{2/3} \sigma^{-2/3} \log^{1/3}(|\mathcal{H}|T)$ finishes the proof. $\qquad \square$

## C   Proofs in Section 3.3

**Theorem 2.** *Under Assumptions 1-5, Algorithm 2 with certain choice of $N$ and $\gamma$ guarantees that $\mathbf{Reg}_{\mathsf{CB}} = \mathcal{O}\left( \sqrt{KT\mathbf{Reg}_{\mathsf{Sq}}} + \sigma^{-2/3}(KT)^{2/3} \log^{1/3}(|\mathcal{H}|T) \right)$.*

*Proof.* According to Eq. (9), we know that with probability at least $1 - \delta$,

$$\sum_{a=1}^{K} \mathbb{E}_{x \sim \mathcal{D}} \mathbb{E}_{y|x,a} \left( \widehat{h}_a(x,y) - h_a^{\star}(x,y) \right)^2$$

$$\leq K \cdot \mathbb{E}_{x \sim \mathcal{D}, a \sim \pi_{\text{Unif}}, y|x,a} \left[ \left( \widehat{h}_a(x,y) - h_a^{\star}(x,y) \right)^2 \right]$$

$$\leq K \cdot \mathbb{E}_{x \sim \mathcal{D}, a \sim \pi_{\text{Unif}}, y|x,a} \left[ \|\widehat{h}(x,y) - h^{\star}(x,y)\|_2^2 \right]$$

$$\leq \mathcal{O} \left( \frac{K \log \frac{|\mathcal{H}|}{\delta}}{N} \right), \tag{18}$$

where the last inequality uses Eq. (9). Let $a_t^{\star} = \arg\max_a f^{\star}(x_t, a)$ and recall that $\underline{f}^{\star}(x,a) := \mathbb{E}_{y|x,a} \left[ G(h_a^{\star}(x,y), \frac{\theta}{\alpha} - \sigma, \sigma) \right] \in \mathcal{F}$. We have

$$\mathbf{Reg}_{\mathsf{CB}} = \mathbb{E} \left[ \sum_{t=1}^{T} f^{\star}(x_t, a_t^{\star}) - \sum_{t=1}^{T} f^{\star}(x_t, a_t) \right]$$

$$= \mathbb{E} \left[ \sum_{t=1}^{T} \mathbb{E}_{y|x_t, a_t^{\star}}[\phi^{\star}(x_t, y)] - \sum_{t=1}^{T} \mathbb{E}_{y'|x_t, a_t}[\phi^{\star}(x_t, y')] \right] \qquad \text{(From Assumption 2)}$$

$$\leq \mathbb{E} \left[ \sum_{t=1}^{T} \underline{f}^{\star}(x_t, a_t^{\star}) - \sum_{t=1}^{T} \underline{f}^{\star}(x_t, a_t) \right] \qquad \text{(From Lemma 4)}$$

$$\leq \frac{\gamma}{4} \mathbb{E} \left[ \sum_{t=1}^{T} (\widehat{f}_t(x_t, a_t) - \underline{f}^{\star}(x_t, a_t))^2 \right] + \mathcal{O} \left( \frac{TK}{\gamma} \right) \tag{19}$$

The last step is from Foster and Rakhlin [2020, Lemma 3]. Recall that $\widehat{r}_\sigma(x_t, y_t, a_t) = G(\widehat{h}_{a_t}(x_t, y_t), \frac{\theta}{\alpha} - \sigma, \sigma)$. Direct calculation shows that

$$\mathbf{Reg}_{\mathsf{Sq}} \geq \mathbb{E} \left[ \sum_{t=1}^{T} (\widehat{f}_t(x_t, a_t) - \widehat{r}_\sigma(x_t, y_t, a_t))^2 - \sum_{t=1}^{T} (\underline{f}^{\star}(x_t, a_t) - \widehat{r}_\sigma(x_t, y_t, a_t))^2 \right]$$

$$= \mathbb{E} \left[ \sum_{t=1}^{T} (\widehat{f}_t(a_t) - \underline{f}^{\star}(x_t, a_t))(\widehat{f}_t(x_t, a_t) + \underline{f}^{\star}(x_t, a_t) - 2\widehat{r}_\sigma(x_t, y_t, a_t)) \right]$$

$$= \mathbb{E} \left[ \sum_{t=1}^{T} (\widehat{f}_t(x_t, a_t) - \underline{f}^{\star}(x_t, a_t))^2 \right]$$

$$+ 2\mathbb{E} \left[ \sum_{t=1}^{T} (\widehat{f}_t(x_t, a_t) - \underline{f}^{\star}(x_t, a_t))(\underline{f}^{\star}(x_t, a_t) - \mathbb{E}_{y_t}[\widehat{r}_\sigma(x_t, y_t, a_t)]) \right]$$

$$\geq \frac{1}{2} \mathbb{E} \left[ \sum_{t=1}^{T} (\widehat{f}_t(x_t, a_t) - \underline{f}^{\star}(x_t, a_t))^2 \right] - 2\mathbb{E} \left[ \sum_{t=1}^{T} (\underline{f}^{\star}(x_t, a_t) - \mathbb{E}_{y_t}[\widehat{r}_\sigma(x_t, y_t, a_t)])^2 \right], \tag{20}$$

where the last step is by AM-GM inequality. For the second term, we know that

$$\mathbb{E} \left[ \sum_{t=1}^{T} \left( \underline{f}^{\star}(x_t, a_t) - \mathbb{E}_{y_t}[\widehat{r}_\sigma(x_t, y_t, a_t)] \right)^2 \right]$$

$$= \mathbb{E} \left[ \sum_{t=1}^{T} \left( \mathbb{E}_{y_t|x_t, a_t} \left[ G\left( h_{a_t}^{\star}(x_t, y_t), \frac{\theta}{\alpha} - \sigma, \sigma \right) - G\left( \widehat{h}_{a_t}(x_t, y_t), \frac{\theta}{\alpha} - \sigma, \sigma \right) \right] \right)^2 \right]$$

$$\leq \mathbb{E} \left[ \sum_{t=1}^{T} \mathbb{E}_{y_t|x_t, a_t} \left[ \left( G\left( h_{a_t}^{\star}(x_t, y_t), \frac{\theta}{\alpha} - \sigma, \sigma \right) - G\left( \widehat{h}_{a_t}(x_t, y_t), \frac{\theta}{\alpha} - \sigma, \sigma \right) \right)^2 \right] \right]$$

$$\leq \mathbb{E}\left[\sum_{t=1}^{T}\sum_{a=1}^{K}\mathbb{E}_{y|x_t,a}\left[\left(G\left(h_a^\star(x_t,y),\frac{\theta}{\alpha}-\sigma,\sigma\right)-G\left(\widehat{h}_a(x_t,y),\frac{\theta}{\alpha}-\sigma,\sigma\right)\right)^2\right]\right]$$

$$\leq \frac{1}{\sigma^2}\mathbb{E}\left[\sum_{t=1}^{T}\sum_{a=1}^{K}\mathbb{E}_{y|x_t,a}\left[\left(h_a^\star(x_t,y)-\widehat{h}_a(x_t,y)\right)^2\right]\right]$$

$$\leq \mathcal{O}\left(\frac{TK\log(|\mathcal{H}|T)}{\sigma^2 N}\right), \tag{21}$$

where the first inequality is from Jensen's inequality; the third inequality is from that $G(v,\beta,\sigma)$ is $\frac{1}{\sigma}$-Lipschitz in $v$; and the last inequality is by Eq. (18) with $\delta=\frac{1}{T}$. Combining Eqs. (19)-(21), we obtain that

$$\mathbf{Reg}_{\mathsf{CB}} \leq \mathcal{O}\left(\gamma\mathbf{Reg}_{\mathsf{Sq}}+\frac{TK\gamma\log(|\mathcal{H}|T)}{\sigma^2 N}+\frac{TK}{\gamma}+N\right).$$

Picking $N=\frac{1}{\sigma}\sqrt{TK\gamma\log(|\mathcal{H}|T)}$ and $\gamma=\min\left\{\sqrt{KT/\mathbf{Reg}_{\mathsf{Sq}}},\sigma^{-2/3}(KT)^{2/3}\log^{-1/3}(|\mathcal{H}|T)\right\}$, we obtain that

$$\mathbf{Reg}_{\mathsf{CB}} \leq \mathcal{O}\left(\sqrt{KT\mathbf{Reg}_{\mathsf{Sq}}}+\sigma^{-2/3}(KT)^{2/3}\log^{1/3}(|\mathcal{H}|T)\right),$$

which finishes the proof. $\square$

## D  Omitted Details in Section 4

### D.1  Prompt Used in Generating Dataset

The prompt we use in answer generation basically the question itself shown as follows. We replace "question" by $x$ in our experiment.

> **Prompt**
>
> {question}

The prompt we use in feedback generation as follows. We replace "question" and "answer" by $x$ and $a$ respectively in our experiments. We replace "mood" by "satisfied" ("not satisfied") when the answer is generated by Qwen1.5-32B-Chat (Qwen1.5-0.5B-Chat).

> **Prompt**
>
> System: You are a user simulator. A user has been presented a Question and an Answer. Simulate the user's next statement. The user is {mood} with the Answer to the Question.
> Question: {question}
> Answer:{answer}

### D.2  Prompt Used in Learning Inverse Kinematic Model

The prompt we use in learning the inverse kinematic model is as follows. We replace "question", "answer", and "feedback" by $x$, $a$, and $y$ respectively in our experiments.

> **Prompt**
>
> System: You are a conversation evaluating agent. Given a User's Question, an Answer, and the User's Feedback: determine if the User's Feedback is consistent with Answer. Respond with Yes or No only.
> User's Question: {question}
> Answer:{answer}
> User's Feedback:{feedback}

> Respond with Yes or No only.

## D.3  Prompt Used in Learning Policy

The prompt we use in learning the reward predictor is as follows. We replace "question" and "answer" by $x$ and $a$ respectively in our experiments.

**Prompt**

System: You are an Answer evaluating agent. Given a User's Question and an Answer: assess if the Answer is good. Respond with Yes or No only.
User's Question:{question}
Answer:{answer}
Respond with Yes or No only.

