# OpenReview forum: "Provably Efficient Interactive-Grounded Learning with Personalized Reward"
_NeurIPS.cc/2024/Conference — NeurIPS 2024 poster_

### Official Review · Reviewer_TjTV · 2024-06-17

**Soundness:** 2
**Presentation:** 3
**Contribution:** 2
**Rating:** 5
**Confidence:** 3

**Summary:**

This paper considers Interactive-Grounded Learning (IGL) with personalized reward, where the feedback can be context-dependent. The authors propose two algorithms, which are provably efficient by utilizing the novel Lipschitz reward estimators. Empirical results on the image classification dataset and the conversational dataset showcase the effectiveness of the proposed algorithms.

**Strengths:**

(+) This is the first work to provide provably efficient algorithms with regret guarantees for IGL with personalized reward.

(+) It is interesting to see the experiments on the conversation dataset, which is a novel task in the IGL literature.

**Weaknesses:**

(-) The paper does not compare the proposed algorithms to the IGL-P algorithm proposed in (Maghakian et al., 2022).

(-) While there are justifications in the paper for using the Lipschitz reward estimators (Lemma 4), it is unclear why using $\mathbb{1}\\{\hat{h}_a(x,y)\ge\frac{\theta}{\alpha}\\}$ or the previous IGL-P algorithm could fail.

**Questions:**

1. It would be nice to provide counter-examples to show why using $\mathbb{1}\\{\hat{h}_a(x,y)\ge\frac{\theta}{\alpha}\\}$ or the step-function estimator in prior work could fail in the setting.
2. How restrictive is it to obtain prior knowledge of parameters $\alpha$ and $\theta$ in Algorithms 1 and 2? What is the role of parameter $\gamma$ in Algorithm 2?

**Limitations:**

No "Limitations" section is provided in the paper.

---

> ### Author Rebuttal · Authors · 2024-08-07
>
> > **“The paper does not compare the proposed algorithms to the IGL-P algorithm proposed in (Maghakian et al., 2022).”**
>
> Reply: We followed the reviewer’s suggestion and tested the algorithm of Maghakian et al. on the MNIST dataset. We found that it achieves less than 0.2 average progressive reward, significantly worse than our algorithms. We will add this result to our paper in the next version.
>
> Beside the issue of using step-function estimators, another main reason for this huge performance difference is that IGL-P trains the reward predictor (inverse kinematics part) in an online manner. Specifically, in each iteration, IGL-P updates the predictor using only the current sample, without reusing past samples for training. In contrast, our algorithms first employ uniform policies to collect a dataset and then train the predictor on this dataset for multiple epochs via supervised learning. Since the actions in our dataset are uniformly distributed, the dataset has good coverage for each action, ensuring better generalization performance for our predictor.
>
> > **“While there are justifications in the paper for using the Lipschitz reward estimators (Lemma 4), it is unclear why using $\mathbb{1}\lbrace\hat h_a(x,y)\ge\frac{\theta}{\alpha}\rbrace$ or the previous IGL-P algorithm could fail.”**
>
> Reply: We do not have counter-examples showing that the binary reward function provably fails. However, from a technical perspective, Lipschitzness is crucial to control the difference between the empirical value function with respect to true rewards and that with respect to the estimated rewards (lines 486-487), and from an empirical perspective, we also show that our algorithm performs much better when equipped with our Lipschitz reward estimator compared to the binary one.
>
> To further demonstrate the last point, we run additional experiments using the on-policy Algorithm 2 on MNIST with 20 different random seeds. The binary reward estimator achieves an average progressive reward of 0.711 (0.040) and a test accuracy of 88.5% (3.5%). In contrast, the Lipschitz reward estimator achieves an average progressive reward of 0.748 (0.025) and a test accuracy of 90.6% (3.4%). According to the two-sample t-test, the Lipschitz reward estimator outperforms the binary one with greater than 95% confidence in terms of both average progressive reward and test accuracy.
>
> > **“How restrictive is it to obtain prior knowledge of parameters $\alpha$ and $\theta$ in Algorithms 1 and 2? What is the role of parameter $\gamma$ in Algorithm 2?”**
>
> Reply: We think prior knowledge on $\alpha$ and $\theta$ is usually easy to obtain in many setups. For example, as we mentioned after Assumption 3, in the s-multi-label classification problem (with $1\leq  s < K/2$), we have $\alpha = s$ and $\theta = 1$.
>
> The parameter $\gamma$ controls the amount of exploration in the inverse-gap weighting (IGW) rule. Intuitively, smaller $\gamma$ leads to more exploration among the actions.

---

> > ### Comment · Reviewer_TjTV · 2024-08-11
> > **Follow-up Response**
> >
> > I thank the authors for their responses to my questions. I appreciate that they include numerical results on the comparison with the IGL-P algorithm. Responding:
> >
> > **Q1.** For interpretability, in the later version of the paper, I would suggest plotting the empirical errors of $\mathbb{1}\{\hat{h}_a(x,y)\geq\frac{\theta}{\alpha}\}$ and $G(\hat{h}_a(x,y),\frac{\theta}{\alpha}-\sigma,\sigma)$ compared to the ground-truth estimator $\mathbb{1}\{h_a^\star(x,y)\geq\frac{\theta}{\alpha}\}$.

---

> > > ### Author Response · Authors · 2024-08-11
> > >
> > > Thanks for your suggestion. We will incorporate this into the next version. If our response addresses your concern, please do consider re-evaluating our paper.

---

### Official Review · Reviewer_LmpU · 2024-07-10

**Soundness:** 2
**Presentation:** 2
**Contribution:** 2
**Rating:** 5
**Confidence:** 3

**Summary:**

This paper studies the problem of personalized rewards in the context of Interactive-Grounded Learning (IGL), where the goal is to maximize the unobservable latent rewards from the observed reward-dependent feedback on actions being taken. Specially, authors introduce provably efficient algorithms with sublinear regret to solve a variant of IGL, in which the feedback depends on both the context and rewards. Authors introduce Lipschitz reward estimator via inverse kinematics. Based on it, two algorithms are proposed based on explore-then-exploit and inverse-gap weighting respectively. Both achieves $O(T^{2/3})$ regret. Empirical studies are performed on an image classification dataset and a conversational dataset.

**Strengths:**

This paper explores a variant setting of IGL. Instead of sticking with the conditional independence assumption made in existing works, authors go one-step further by studying the setting where the observed feedback depends on both rewards and the context. This setting is practical in cases such as recommender systems.

The authors introduces two algorithms that enjoy the sublinear regret of $\tilde{O}(T^{2/3})$ for IGL with personalized reward. In particular, a new reward estimator is introduced via inverse kinematics to construct Lipschitz rewards. Compared to the prior work (Maghakian et al., 2022) which studies deterministic binary reward, here the reward estimator generalizes to randomized binary rewards.

**Weaknesses:**

While this paper clearly articulates the idea, my main concern lies in its technical novelty and contribution. More specifically,

1. While I understand the motivation of the paper, It is not clear to me how the studied setting can be technically more challenging than the common contextual bandits. The studied setting concerns about partial feedback that depends on latent reward and context, whereas contextual bandits concerns about explicit rewards that depend on context and action. The studied setting is somewhat a simplified version of POMDP.  As such, extending the stochastic contextual bandit algorithms and their theoretical guarantees for IGL appears to be straightforward.

2. Prior study of Xie et al. [2022] has relaxed the conditional independent assumption to study feedback which depends on both action and reward. A natural extension will be to deal with feedback that depends on context, action, and reward (i.e. $y|x, a, r$) when it comes to the personalized settings. It is also more practical and align with the contextual bandit settings. However, authors preserves the conditional independent assumption for actions.

3. The main contribution and emphasis lies in the construction of the reward estimator, the novelty in algorithmic design appears to be limited by using simple standard bandit algorithms (e.g. explore-then-exploit). In addition, algorithmically, employing uniform exploration may lead to higher sample complexity for the balance between exploration and exploitation. More advanced exploration strategies might need to be considered. As such, authors are suggested to comment on the optimality of the provided regret bound.

4. Compared to Algorithm 1, the performance guarantee of Algorithm 2 relies on more restricted assumptions (Assumptions 1 - 5), which can be difficult to satisfied.

**Questions:**

1. Could you explain why do we need an underestimate of the reward in this studied setting? Most of the time when we study online settings, optimism is desirable.

2. What is the intuition of $\sigma$?

3. In Table 1, why the performance of algorithm 2 outperforms algorithm 1? Do they converge to similar performance if running for longer time horizon? It is suggested to provide the regret plots for experiments.

**Limitations:**

No potential negative social impact.

---

> ### Author Rebuttal · Authors · 2024-08-07
>
> > **Comparison with contextual bandits and POMDP**
>
> As the reviewer already pointed out, in IGL one only observes indirect feedback about the reward, while in standard contextual bandits one observes the reward directly. This clearly makes IGL more challenging than contextual bandits — in fact, without further assumptions (like those we made), learning in IGL becomes impossible. So we do not understand why the reviewer thinks IGL is not technically more challenging. If what you mean is that all we need is to construct reward estimators using the feedback received, then we emphasize again that our Lipschitz reward estimator is novel, and the previous estimator proposed by Maghakian et al. (2022) does not lead to a regret guarantee.
>
> Also, we emphasize that IGL is not a simplified version of POMDPs, and they present their own unique challenges. In POMDPs, although the latent state is unobservable, the reward of the chosen action is still revealed to the learner, which means that POMDP algorithms do not need to address the symmetry breaking problem that is central to IGL.
> > **The conditional independence assumption for actions**
>
> Note that [Xie et al., 2022] in fact considers the conditional independence assumption for context, while we follow [Maghakian et al. (2022)] that considers the conditional independence assumption for actions. The two main reasons we study this setting are:. First, we believe that it does capture many real-world applications. For instance, in recommender systems, while different users may communicate their feedback differently, a given user typically expresses (dis)satisfaction in a consistent way. Thus, this is a realistic setting where the feedback depends only on the context (the user) and the reward (degree of satisfaction), but not the actual action (the recommended item).
>
> Second, this seemingly restricted setting is already challenging enough, and there are no efficient algorithms with sublinear regret before our work. Thus, we believe that solving this challenge (which we did) is itself a significant contribution. Furthermore, our empirical results also validate the practicality and the effectiveness of our approach.
>
> We do recognize the limit of our setting as pointed out by the reviewer though, and we leave the more general setting as an important future direction.
> > **“Require more advanced exploration strategies. Optimality of the provided bound.”**
>
> Note that in addition to the simple explore-then-exploit idea used in Algorithm 1, we also use the more sophisticated online exploration idea from Foster and Rakhlin [2020] in Algorithm 2. While we are not able to prove a better regret bound (mainly due to the fact that for learning $\hat{h}$, we still use uniform exploration), Algorithm 2 does perform better than Algorithm 1 empirically, as shown in our experiments.
>
> We are not certain about the optimality of our $O(T^{2/3})$ regret bounds. An obvious regret lower bound is $\Omega(\sqrt{KT})$ (from contextual bandits). Given that IGL is significantly more challenging (and also that $T^{2/3}$ is the optimal regret for a class of partial monitoring problems as mentioned by Reviewer 8SUA), it is possible that $T^{2/3}$ regret is indeed optimal in our setting.
> > **“Compared to Algorithm 1, Algorithm 2 relies on more assumptions”**
>
> We would like to clarify that Algorithm 2 only relies on two additional assumptions, Assumptions 4 and 5, both of which are commonly used in the contextual bandit literature and reasonable in practice.
>
> Assumption 4 assumes that the square loss regret of the online oracle is bounded. As stated in line 240, we can use Vovk’s aggregation algorithm to achieve logarithmic regret for finite $\mathcal{F}$. Additionally, as noted in footnote 1, there are many examples of regression oracles when $\mathcal{F}$ is not finite, and we refer the reviewer to Foster and Rakhlin [2020] for further details.
>
> Assumption 5 assumes that our function class realizes one specific function $\underline{f}^\star$. This is similar to the widely used realizability assumption and can be satisfied with a rich function class such as deep neural networks.
> > **“Why a reward underestimator?”**
>
> Indeed, in many bandit problems, optimism is a commonly used exploration strategy. The fact that we need pessimism instead highlights the difference between IGL and standard contextual bandits. From a technical viewpoint, the reason we make sure that the predicted rewards are underestimators and at the same time pretty accurate for the optimal policy (Lemma 4) is that it then allows us to upper bound the regret under the true reward by the regret under the predicted reward (which the algorithm tries to minimize).
> > **“Intuition of $\sigma$?”**
>
> Intuitively, $\sigma$ characterizes the difference of function $h^\star_{\pi^\star(x)}(x,y)$ when $\phi(x,y)=0$ and $\phi(x,y)=1$. As discussed in line 145-154, when $\phi(x,y)=0$, we have $h^\star_{\pi^\star(x)}(x,y)\leq \frac{1}{K-\alpha}$ and when $\phi(x,y)=1$, we have $h^\star_{\pi^\star(x)}(x,y)\geq \frac{\theta}{\alpha}$. $\sigma$ is then set to be half of this gap when constructing our reward estimator that is $1/\sigma$-Lipschitz.
> > **“why algorithm 2 outperforms algorithm 1? Do they converge to similar performance? Regret plots are suggested.”**
>
> Although Algorithm 2 has the same $T^{2/3}$ regret bound as Algorithm 1, as stated in line 243, this is primarily because both need to uniformly explore in the first phase. However, in the second phase, Algorithm 2 is on-policy and employs the inverse-gap weighting strategy, which typically offers better exploration than the explore-then-exploit approach used in Algorithm 1. This is the main reason why Algorithm 2 empirically performs better, regardless how long the time horizon is. To further illustrate this, we have provided the average progressive reward plots in the rebuttal PDF, which indeed demonstrates that Algorithm 2 consistently outperforms Algorithm 1 over time.

---

> ### Comment · Reviewer_LmpU · 2024-08-13
>
> I thank the authors for their detailed response.
>
> To further clarify,  Xie et al. [2022] studies the setting of $(y|a, r)$, the current draft studies the setting of $(y|x, r)$. Could you elaborate more on why the studied setting may not borrow the techniques from Xie et al. [2022]? More specifically, what are the main technical differences when dealing with feedback depending on context and reward v.s. feedback depending on action and reward?

---

> > ### Author Response · Authors · 2024-08-13
> >
> > Thanks for your further questions. We emphasize that the idea behind the algorithm design of Xie et al., [2022] **cannot** be used in our setting. Specifically, Xie et al., [2022] consider the setting where the feedback is independent of the context given the reward and the action, so they learn the value function $f_a$ and the reward decoder $\psi_a$ separately for each action $a\in[K]$. Generalizing their idea to our setting would then mean learning the value function and the reward decoder **for each context**, which is infeasible since there could be infinitely many contexts. Therefore, we have to propose a new method, that is, constructing a reward decoder via inverse kinematic, for our setting.

---

> > > ### Comment · Reviewer_LmpU · 2024-08-13
> > >
> > > Thank you for your further clarification, which does help better understand the context. Please ensure to address the original concerns and incorporate the feedback in revision. The clear explanations will enhance the clarity of the current draft.
> > >
> > > Considering the responses from authors during rebuttal, I raise my score accordingly.

---

### Official Review · Reviewer_qSGo · 2024-07-13

**Soundness:** 3
**Presentation:** 2
**Contribution:** 2
**Rating:** 5
**Confidence:** 2

**Summary:**

In this work, the authors provide the first provably efficient algorithms with sublinear regret guarantees for Interactive-Grounded Learning (IGL) with personalized rewards under realizability. Based on a novel Lipschitz reward estimator, the authors propose two algorithms: one based on explore-then-exploit and the other based on inverse-gap weighting. Furthermore, the authors apply IGL to learning from image feedback and learning from text feedback, showcasing the effectiveness of the proposed algorithms.

**Strengths:**

This work provides the first provably efficient algorithms for Interactive-Grounded Learning with personalized rewards, supported by experiments demonstrating the effectiveness of the proposed algorithms.

The presentation is generally clear, with detailed experiment settings, and the appendix is well-organized. The definitions and assumptions are well defined.

**Weaknesses:**

The theoretical contributions are limited by the realizability and identifiability assumptions. It would be helpful to provide practical examples where all these assumptions are satisfied.

In Table 1, the difference between Binary and Lipschitz reward estimation seems insignificant, which raises doubts about the practical contribution of the new estimator.

**Questions:**

Same as weakness.

Q1: It seems straightforward to apply explore-then-exploit and inverse-gap weighting. Could you please state the challenges encountered when proving the regret guarantees?

Q2: It would be helpful if the authors could provide some related lower bounds to help readers understand the gap between the current regret bound and the optimal rate.

**Limitations:**

Same as weakness.

---

> ### Author Rebuttal · Authors · 2024-08-07
>
> > **“The theoretical contributions are limited by the realizability and identifiability assumptions. It would be helpful to provide practical examples where all these assumptions are satisfied.”**
>
> Reply: Realizability is a well-established assumption in the contextual bandit literature [Foster et al., 2018, Foster and Rakhlin, 2020], and can be satisfied by using a rich function class as the reward predictor such as deep neural networks.
>
> Identifiability is a necessary assumption in IGL to break the symmetry between reward being 1 and being 0. As demonstrated by examples in [Xie et al. ,2022], learning in IGL is impossible without breaking the reward symmetry. As mentioned in Lines 107-108, our identifiability assumption is satisfied in many scenarios with sparse rewards, including classification problems and recommender systems where the user is primarily interested in a very small subset of items.
>
>
> > **“In Table 1, the difference between Binary and Lipschitz reward estimation seems insignificant, which raises doubts about the practical contribution of the new estimator.”**
>
> Reply: To verify that the advantage of our Lipschitz reward estimator is indeed significant, we run additional experiments using the on-policy Algorithm 2 on MNIST with 20 different random seeds. The binary reward estimator achieves an average progressive reward of 0.711 (0.040) and a test accuracy of 88.5% (3.5%). In contrast, the Lipschitz reward estimator achieves an average progressive reward of 0.748 (0.025) and a test accuracy of 90.6% (3.4%).
>
> According to the two-sample t-test, our Lipschitz reward estimator outperforms the binary one with greater than 95% confidence in terms of both average progressive reward and test accuracy. These results demonstrate the practical significance and contribution of the new estimator.
>
> > **“It seems straightforward to apply explore-then-exploit and inverse-gap weighting. Could you please state the challenges encountered when proving the regret guarantees?”**
>
> Reply: While explore-then-exploit and inverse-gap weighting are well-established techniques in contextual bandits where the learner has access to the true reward, applying these methods in the context of IGL, where the learner only receives certain indirect feedback, presents significant challenges.
>
> First, to develop a provable algorithm, it is necessary to design an effective reward predictor. The step-function estimator used by Maghakian et al. (2022) is one such example. However, our analysis indicates that this estimator does not lead to a regret guarantee. Instead, we propose a novel Lipschitz reward estimator, which is essential for proving the regret guarantees.
>
> Additionally, unlike the widely used optimism principle in bandit problems, we apply a pessimism approach instead: our predicted reward is an underestimator of the true reward. This design is also crucial for obtaining the regret guarantee, whose analysis is nontrivial: it involves leveraging the Lipschitz property of our estimator to address the discrepancy between the predicted reward and the true reward. This differs from the misspecification analysis typically conducted in contextual bandits.
>
> To sum up, our approach and the theoretical analysis address the unique challenges posed by the IGL problem and are innovative and nontrivial in our opinion.
>
> > **“It would be helpful if the authors could provide some related lower bounds to help readers understand the gap between the current regret bound and the optimal rate.”**
>
> Reply: According to the lower bound in the contextual bandit problem, the best regret bound we can hope for in IGL is $O(\sqrt{KT})$. However, as we stated before, IGL is substantially more difficult than contextual bandit, so whether $O(\sqrt{T})$ regret is achievable remains an open question. Nevertheless, we emphasize that our work presents the first provably efficient algorithm with sublinear regret in the personalized reward setting.

---

### Official Review · Reviewer_8SUA · 2024-07-17

**Soundness:** 3
**Presentation:** 4
**Contribution:** 3
**Rating:** 7
**Confidence:** 4

**Summary:**

The authors provide sublinear regret algorithms for Interaction Grounded Learning (IGL) setting, a modification of the standard contextual bandit setting where instead of getting the reward signal, the learner receives some alternative signal from an arbitrary space. The game proceeds for T rounds where at every round, the learner gets a context x_t, chooses an action a_t, gets a reward r(x_t, a_t) and receives feedback y_t \in Y.

The primary difference from the prior work is that the reward function can depend on the context x, i.e. the reward function can be personalized for the context x. Due to this personalization, the authors make an independent assumption that given the reward and the context, the observation and the action are independent to each other.

**Strengths:**

Provide T^{2/3} style of regret bounds for IGL setting with personalized rewards.

Most of the assumptions and problem setting follow classical contextual bandits literature.

Experimental are provided for the corresponding algorithms.

**Weaknesses:**

The regret bound scales as T^{2/3} and it is not clear how or when can we get T^{1/2}.  I am guessing that similar to the partial monitoring setting we will need to make assumptions on the function \Phi and there will be a dichotomy between T^{1/2} regret vs T^{2/3} regret based on the information structure.

The proof techniques follow closely well-known tools like inverse kinematics and SquareCB from the contextual bandits and RL literature. However, the experiments are still quite interesting.

**Questions:**

Can the authors provide a more detailed discussion of how IGL is different from the partial monitoring setting. From what I understand, the primary difference is the assumption that learner has access to a class \Phi which contains a decoder \phi that maps the context and observation to actions. Thus this becomes closed to Model-Based methods in RL, etc, and hence we can expect statistical traceability. Can the authors elaborate if there are any other differences?

Paper is well written.

---

> ### Author Rebuttal · Authors · 2024-08-07
>
> > **“Can the authors provide a more detailed discussion of how IGL is different from the partial monitoring setting. From what I understand, the primary difference is the assumption that the learner has access to a class $\Phi$ which contains a decoder $\phi$ that maps the context and observation to actions. Thus this becomes close to Model-Based methods in RL, etc, and hence we can expect statistical traceability. Can the authors elaborate if there are any other differences?”**
>
> Reply: Besides the differences mentioned by the reviewer, in the partial monitoring setting, in order to achieve sublinear regret guarantees, it is assumed that the difference between the loss of any two arms can be represented as a certain linear combination of the signal matrix. Comparing the assumptions in these two different problems, we think those in IGL might be easier to interpret in real applications.

---

### Author Rebuttal · Authors · 2024-08-07

We thank all the reviewers for their detailed and valuable comments. To further illustrate that Algorithm 2 outperforms Algorithm 1, as suggested by Reviewer LmpU, we include the average progressive reward plot in the rebuttal PDF, which indeed demonstrates that Algorithm 2 consistently outperforms Algorithm 1 over time.

---

### Decision · Program_Chairs · 2024-09-25

**Decision:**

Accept (poster)

**Comment:**

All the reviewers have a positive feedback on this paper concerning IGL. It seems that the authors received several bits of constructive feedback in the discussion phase, much of which can be easily incorporated through discussion of the setting, results and better contextualization of their technical challenges and accomplishments. I encourage the authors to address these concerns.